# Human intestinal nematode infections in Sri Lanka: A scoping review

**Nalini Kaushalya Jayakody**[1,2]*, **Anjana Silva**[2], **Susiji Wickramasinghe**[3], **Nilanthi de Silva**[4], **Sisira Siribaddana**[5], **Kosala Gayan Weerakoon**[2]*

**1** Department of Parasitology, Faculty of Medicine, Wayamba University of Sri Lanka, Kuliyapitiya, Sri Lanka, **2** Department of Parasitology, Faculty of Medicine and Allied Sciences, Rajarata University of Sri Lanka, Saliyapura, Sri Lanka, **3** Department of Parasitology, Faculty of Medicine, University of Peradeniya, Kandy, Sri Lanka, **4** Department of Parasitology, Faculty of Medicine, University of Kelaniya, Ragama, Sri Lanka, **5** Department of Medicine, Faculty of Medicine and Allied Sciences, Rajarata University of Sri Lanka, Saliyapura, Sri Lanka

* jayakodynalini@wyb.ac.lk (NKJ); kosalagadw83@gmail.com (KGW)

**Data Availability Statement:** All the data included in the review is provided within the manuscript and in supporting information.

## Abstract

### Background

Sri Lanka, an island located in South Asia, once experienced a notable prevalence of human intestinal nematode infections (HINIs). With the implementation of control programs, infection prevalence was reduced. Detailed information on prevalence, distribution and temporal trends of HINIs is limited. This review aims to explore Sri Lanka's HINI distribution, trends, diagnostics, control and challenges.

### Methodology

We reviewed published information on HINIs in Sri Lanka in electronic databases, local journals and grey literature from inception to September 2022. Based on the Preferred Reporting Items for Systematic Reviews and Meta-Analyses extension for Scoping Reviews (PRISMA-Scr), a systematic strategy was used for searching, screening, reviewing and data extraction. The screening was initiated with a review of titles and abstracts using specific keywords, followed by a full-text screening based on predefined eligibility criteria. A total of 105 studies were included in the review, with 28 selected for pooled prevalence analysis.

### Principal findings

The first nationwide survey in 1924 reported a hookworm infection prevalence of 93.1%. By 2017, soil-transmitted helminth (STH) infection prevalence across the island was 0.97% (ascariasis-0.45%, trichuriasis-0.25%, and hookworm infection-0.29%), and the enterobiasis prevalence between 2003 and 2017 ranged from 0% to 42.5%. Strongyloidiasis had been understudied, with a prevalence of 0.1% to 2%. Over the past two decades, the island-wide pooled HINI prevalence was 13.3%. Within specific demographics, it was 6.96% in the general community, 33.4% in plantation sector, and 11.6% in slum communities. During the colonial period, hookworm infection was the commonest HINI, but ascariasis is now more

**Funding:** This review is part of an intestinal parasitoses community evaluation project funded by the National Research Council of Sri Lanka, grant NRC 20-118 to KGW. https://www.nrc.gov.lk. The funders had no role in study design, data collection and analysis, decision to publish, or preparation of the manuscript.

**Competing interests:** The authors have declared that no competing interests exist.

prevalent. The prevailing data relied solely on microscopy, often utilising single stool smears. Mass deworming programs were widely pursued in the first half of the 20th century, initially targeting antenatal women and schoolchildren, and now focusing on specific community groups. National surveys continue monitoring the three main STH infections.

## Conclusions

The significant reduction in STH prevalence in the country over the past ten decades highlights the effectiveness of public health interventions, particularly mass deworming programs. Despite the success, STH prevalence disparities persist in vulnerable populations like plantation and slum communities, where hygiene and living conditions continue to pose challenges. Reliance on single stool smear microscopy highlights the need for more sensitive diagnostics to better assess infections. Fluctuating enterobiasis prevalence and limited strongyloidiasis data underscore the importance of continued surveillance and targeted interventions for sustained control and eventual elimination. Sri Lanka's experiences and control measures offer valuable insights for low-income countries in South Asia and beyond, particularly in managing HINIs with limited resources.

## Author summary

Human intestinal nematode infections (HINIs) pose a significant public health concern in Sri Lanka, especially among the deprived communities. Most Sri Lankan research information on HINIs over the years has been scattered and inconsistent, and many are not accessible to all. To address this issue, we conducted an extensive review of the literature from inception to September 2022. The review showed the first HINI to be recorded as hookworm infection, while all five HINI species were reported in Sri Lanka. In the early 1900s infection prevalence was over 90%, and a gradual decline in prevalence over the period was noted. Among the many measures taken to reduce the infection rates since the early part of the 20th century are the national deworming guidelines framed and implemented from 2012 onwards. The current islandwide prevalence of soil-transmitted helminth (STH) infection is less than 1%. Prevalence data on enterobiasis and strongyloidiasis are sparse and the current islandwide prevalence of these two species is not known. As all prevalence estimates were made using microscopic techniques, more sensitive diagnostic methods should be employed to understand the true picture of present low prevalence estimates. The burden of disease is highest in the hill country plantation sector and urban slum communities where living conditions and sanitary facilities are poor. The experiences and control measures implemented in Sri Lanka can offer valuable insights to other countries especially when facing limited resources and similar challenges.

## Introduction

Human intestinal nematode infections (HINIs) are caused by parasitic worms that inhabit the human digestive tract. Disadvantaged communities who live in the tropics and subtropics are more susceptible to their infections [1]. Among these nematodes, *Ascaris lumbricoides* (roundworm), *Strongyloides stercoralis* (threadworm), *Necator americanus*, and *Ancylostoma*

*duodenale* (hookworm) have complicated life cycles compared to those of *Enterobius vermicularis* (pinworm) and *Trichuris trichiura* (whipworm). While *S. stercoralis* and *E. vermicularis* can spread directly from person to person, others, like *A. lumbricoides*, *T. trichiura*, *N. americanus*, and *A. duodenale*, need a mandatory soil phase to mature, hence named soil-transmitted helminth (STH) or geohelminths [2]. HINIs are transmitted either by ingestion of eggs or by infective larvae penetrating the bare skin [2]. In the majority, they cause a wide range of non-specific illnesses [3]. Heavy infections are linked to impaired physical and intellectual development, anaemia, and reduced resistance to invading microbes [3]. Children, expectant mothers, farmers and plantation labourers are the groups most at risk of contracting HINIs [4]. The physiological demands for nutrients for growth and development in children and pregnant women make them more susceptible to complications associated with these diseases [4]. As the infections are linked to inadequate access to safe water, low socioeconomic standards, poor personal hygiene and poor sanitation practices, wearing footwear and gloves in outdoor activities, improving the quality of water, sanitation and hygiene (WASH) can be used as effective prevention and control strategies [5]. Periodic mass anthelminthic medication delivery through school-based programmes and clinics is another control measure [2,6,7].

Globally, over 1.5 billion individuals suffer from STH infections [4]. As of the current global situation, approximately 807–1121 million people are infected with roundworms, 604–795 million with whipworms, 576–740 million with hookworms, 200 million with pinworms, and 600 million with threadworms [4,8,9]. Polyparasitism is common in high-endemic areas [10]. In 2012, the World Health Organization (WHO) identified STH infections as the most common neglected tropical disease (NTD) in their roadmap to combat NTDs [11]. This increased awareness helped in achieving many control and preventive goals of the disease. The maximum rates of HINIs are found in Sub-Saharan Africa, East Asia, China and the Americas [4].

Asia is responsible for 67% of the world's prevalence of STH infections [12–14]. India has the highest reported prevalence of STH infections in South Asia, which is 21% [15]. With an overall prevalence of 18%, *A. lumbricoides* was the most common species of STH in South and Southeast Asia, followed by *T. trichiura* (14%) and hookworm (12%) [16]. However, the overall prevalence of STH infections is decreasing throughout Asia [17].

*A. lumbricoides*, *N. americanus* and *T. trichiura* have been identified as the three main STHs in Sri Lanka since the past century [18]. In Sri Lanka, systematic antenatal mebendazole deworming was introduced in the 1980s. Deworming was integrated into school health programs in 1960 [19]. The islandwide cumulative STH prevalence in Sri Lanka is 0.97% in 2017 [20].

In this review, scattered information on HINIs in Sri Lanka was arranged in a sequential order to understand trends of infection with past and current infection prevalences, identify the isolated pockets with high endemicity, and interventions made. Such information has the potential to direct responsible bodies, both locally and globally, to focus their management efforts on highly endemic areas and needy communities when available resources are limited. The findings of this review would assist in planning future research and guide health sector authorities to strategically address the control and prevention of HINIs in a manner that is both cost-effective and targeted.

## History of the disease in Sri Lanka

Sri Lanka was made a Crown colony (Ceylon) in 1815 [21]. British landowners employed labourers from India to fill the labour deficit in the plantation industry. In 1888, HINIs were first recorded in the annual survey of the Principal Civil Medical Officer (PCMO) of Sri Lanka, when 31 individuals with hookworm infection were discovered [22]. All the diagnosed people were immigrant workers. Hookworm disease was responsible for 269 fatalities by 1897 [22].

The employees were exposed to a diverse spectrum of infections and parasitic illnesses particularly due to the severely unhygienic conditions that prevailed on the plantations [23]. Registrar General of Ceylon's annual report in 1891 noted that the disease had been introduced to the island by the Indian Malabar coolies [23]. By 1916, the disease had affected over 90% of the people living in the estates [21]. A hookworm control programme was started in 1916 by the International Health Board (IHB) of the Rockefeller Foundation in the Matale area, which was home to roughly around 10,000 South Indian immigrant workers [21].

## Social and geo-climatic profile of Sri Lanka

Sri Lanka is a South Asian country with a tropical climate. It is located between 5° 55' and 9° 51' north latitude and 79° 42' and 81° 53' east longitude [24]. In the driest regions (northern, northwestern, and southeastern), the mean annual rainfall is less than 900mm, whereas, in the wettest regions (western slopes of the central highlands), it exceeds 5000mm [24,25]. The mean annual temperature in the central highlands, is around 16°C (1900m above mean sea level), while it is around 27°C in the coastal lowlands [24]. The country is geographically divided into three major ecological zones: wet, arid and dry [24,25]. There are nine provinces, Northern (NP), North Central (NCP), North Western (NWP), Eastern (EP), Uva (UP), Central (CP), Southern (SP), Sabaragamuwa (SBP) and Western (WP), and 25 districts. Districts are second-level administrative divisions, while provinces are at the apex level. The population of Sri Lanka was 22.16 million in 2021 [26]. The estimated gross domestic product per person was dollar 3292 in 2022 [26]. About 25.2% of people were estimated to be poor in 2022 [27]. Of the population, 98.2%, 87.6%, and 18.9% are enrolled in primary, secondary, and tertiary education, respectively [28]. The adult literacy rate was 92.3% in 2019 [26]. According to the Demographic Health Survey 2006 and 2020, Sri Lanka has increased access to safe drinking water from 76% to 93.2% [26].

## Methods

This scoping review was developed following the methodological framework proposed by Arksey and O'Malley in 2005 [29], which has been further developed by Levac et al. [30] and the Joanna Briggs Institute (JBI) [31,32]. Accordingly, the review proceeds through the following five steps; Identifying the review question, Identifying relevant studies, Study selection, Charting the data, Collating, summarising and reporting the results. To enhance the methodological rigour and reporting quality of this scoping review, the Preferred Reporting Items for Systematic Reviews and Meta-Analyses extension for Scoping Reviews (PRISMA-ScR) checklist (S1 PRISMA checklist), was also followed [33].

### Identifying the review question

Following an exploratory review of the literature on HINIs in Sri Lanka, the following questions were identified.

1. Details of the studies conducted in Sri Lanka on HINIs; the number of studies, research area explored and extent of the research.

2. For the past two decades, HINI prevalence, geographical location, trends of the infection, the impact of control programmes on the prevalence and delimitations.

3. Research gaps in understanding HINIs in the Sri Lankan context.

We used the population, concept, and context (PCC) format to align the study selection with the research question.

### Identifying relevant studies

A comprehensive search of PubMed, Google Scholar, CINAHL, Trip, Oxford Journal, Taylor & Francis Online, JSTOR, Emerald Insight, Scopus, Cochrane Library and Science Direct databases was carried out independently by NKJ and KGW using the Search:((((((((((((((("ascaris"[All Fields]) OR ("roundworm"[All Fields])) OR ("necator"[All Fields])) OR ("ancylostoma"[All Fields])) OR ("hookworm"[All Fields])) OR ("strongyloides"[All Fields])) OR ("threadworm"[All Fields])) OR ("trichuris"[All Fields])) OR ("whipworm"[All Fields])) OR ("enterobius"[All Fields])) OR ("pinworm"[All Fields])) OR ("soil transmitted helminth"[All Fields])) OR ("intestinal nematodes"[All Fields])) OR ("geohelminth"[All Fields])) OR ("helminth"[All Fields]))) AND ((("sri lanka"[All Fields]) OR ("ceylon"[All Fields])) OR ("srilanka"[All Fields])). Studies published from inception to September 2022 were reviewed with no language restrictions. Duplicates were removed with the Rayyan software [34]. Local journals, Sri Lankan institutional repositories and administrative archives were also searched electronically and manually. The bibliographies, Medical publications relating to Sri Lanka 1811–1976,1981–1988 [35,36], Health in Sri Lanka 1977–1980 [37], Medical literature 1980–2005 and dissertations and theses presented to the Post Graduate Institute of Medicine, Sri Lanka were also searched. Reference lists of all relevant studies were evaluated for additional articles.

### Study selection

Selected articles from the electronic database search and the hand search of the published literature were transferred to 2022 Rayyan (https://www.rayyan.ai) [34], from which duplicates were removed. Two reviewers (NKJ and KGW) independently screened the selected articles, initially for the title and abstract against the eligibility criteria. All the articles with discrepancies were included for full-text screening. Full-text screening of the selected articles was done by the same reviewers independently. Issues arising from the full-text screening were addressed by a third reviewer (NDS) with joint discussions involving NKJ and KGW. Reasons for the exclusion of any full-text source of evidence are reported. The methodology is illustrated in the PRISMA flowchart (Fig 1).

The selection of eligible studies was guided by the PCC framework.
Inclusion criteria

1. Qualitative and quantitative studies containing data on prevalence, clinical presentation, risk factors, diagnostic approaches, treatment, control programmes, and drug efficacy on HINIs in Sri Lanka.

2. Grey literature including primary research studies, conference abstracts, bibliographies, government reports and guidelines.

3. Reviews, reports, commentaries and editorial articles that address the objectives of this scoping review.

4. There were no language restrictions.

5. Studies published up to September 2022 from inception were included, as the final search of the literature was carried out in September 2022.

    Exclusion criteria

1. Studies carried out on the Sri Lankan communities residing in other countries. (As they are not exposed to the same climatic, socioeconomic, WASH conditions as Sri Lankan residents)

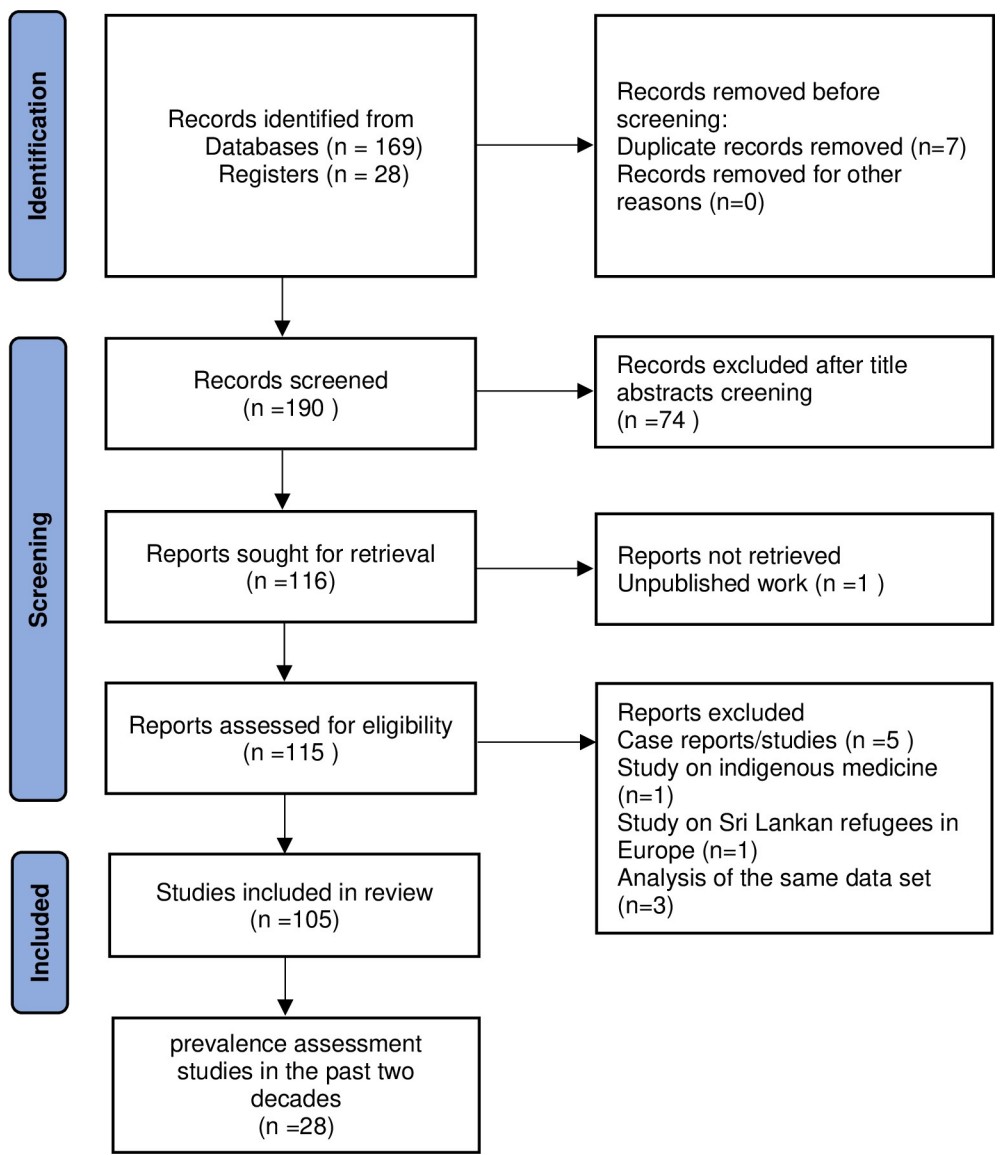

**Fig 1. PRISMA 2020 flow diagram which included searches of databases and registers [38].**

2. Case reports and case series.

3. Veterinary studies.

## Charting the data and quality assessment

The data extraction form was developed in Microsoft Excel using the JBI data extraction instrument as a guide [39]. Two reviewers (NKJ and KGW) used the customised data extraction form to retrieve information independently. The data extraction process was validated on the first ten articles to ensure consistency. The data of the remaining articles were extracted by a single reviewer (NKJ). Questions during data extraction were discussed with KGW and NDS. Data extracted falls into the following domains: publication details, study methodology, study setting, participant information (age, sex, cultural details if available), parasite species,

diagnostic methods, treatments, and control programs carried out, prevalence, intensities of infection, strengths and limitations of the studies. In drug efficacy studies, the name of the drug, dose, dosage, total egg clearance rate (ECR) and egg reduction rate (ERR) were extracted. Extracted data is presented in a summary form ensuring that the objectives of the review are achieved.

Newcastle-Ottawa Quality Assessment (NOQA) Scale modified for cross-sectional studies [40] was utilised to assess the quality of the epidemiological studies for the last two decades (S1 Table). Two reviewers (NKJ and KGW) independently evaluated the quality of the studies and discrepancies that occurred during the process were resolved by the opinion of a third reviewer (NDS). NOQA includes quality assessments in three domains: selection, comparability and outcome. The NOQA tool has scores ranging from 0 to 10. Studies with scores less than 4 were defined as unsatisfactory, 5–6 as satisfactory, 7–8 as good and 9–10 as very good.

## Results

### Studies under review

After the full-text screening, 105 articles were included in the review. All the publications were in the English language. Over 75% (n = 89, 84.7%) of the articles were published since 1975 (Fig 2A). The majority of studies 47 (44.8%) had an epidemiological focus. Others were on drug efficacy, history of the disease, clinical features, diagnosis, treatment, knowledge and associated factors, general reviews, short reports and deworming guidelines (Fig 2B).

### Studies on the efficacy and effectiveness of anthelmintic drugs

Fifteen studies have focused on HINI treatment efficacy and effectiveness (Table 1). The effectiveness of a drug relates to its performance in practical application, contrasting with efficacy, which measures its performance in randomised controlled trials (RCTs) or laboratory studies. In Sri Lanka, all the drug efficacy studies were carried out as RCTs. Parasite detection in pre and post-treatment stool samples was accomplished using direct wet smear (DWS) (50%), Stoll's egg counting (16.6%), Kato-Katz (KK) (33.3%), merthiolate iodine formaldehyde concentration (MIFC) (8.3%), flotation technique (8.3%) and Scotch tape method (8.3%). Two studies have used more than one diagnostic technique for the detection of ova. The efficacy of ten different medications was evaluated in these trials by ECR and ERR. The percentage of individuals who turned stool-negative after treatment is defined as ECR, and the percentage reduction of the mean post-treatment egg count compared to the mean pre-treatment egg count is defined as ERR [41]. Bephenium hydroxynaphthoate (BH), pyrantel pamoate (PP), and tetrachlorethylene (TCE) were effective in treating HINIs [42]. Three daily doses of BH or PP had comparable effects to a single dose of TCE, though both BH and PP are inferior to TCE in a single dose [42–44]. Mebendazole and albendazole are both effective against HINIs, with mebendazole more successful against whipworm and albendazole against hookworm infestations [45–47]. Mebendazole is recommended for multiple infections, especially with whipworm [48], while albendazole is preferred for hookworm-related complications [47]. As mebendazole is cheaper than albendazole, it was recommended for mass drug administration (MDA). Multiple-dose mebendazole regimens are more effective, particularly in its polymorph C form [49,50]. Flubendazole, piperazine, pyrantel and oxantel are effective against ascariasis but less so against trichuriasis [51–53]. Albendazole with diethylcarbamazine (DEC) or alone shows comparable efficacy, while albendazole combined with ivermectin is highly effective against trichuriasis [54,55]. Early trials primarily investigated the effectiveness of BH, TCE, pyrantel, and oxantel, whereas recent studies have shown that albendazole and mebendazole exhibit efficacy in treating HINIs (Table 1).

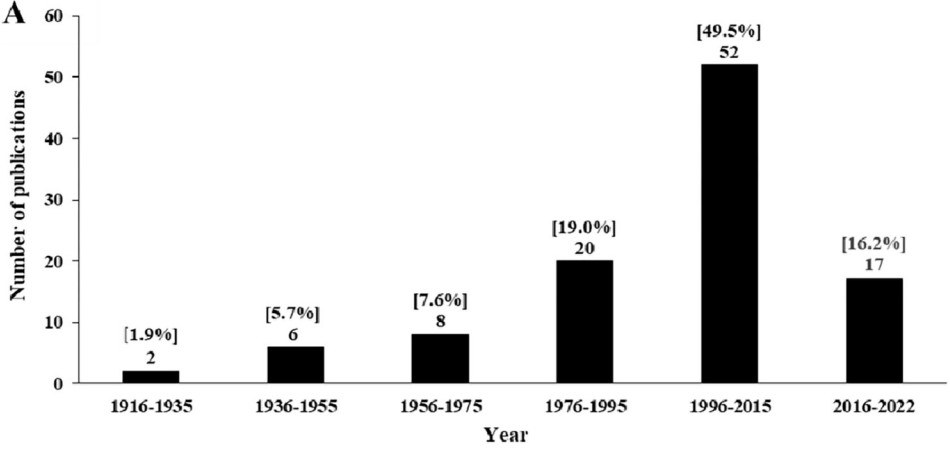

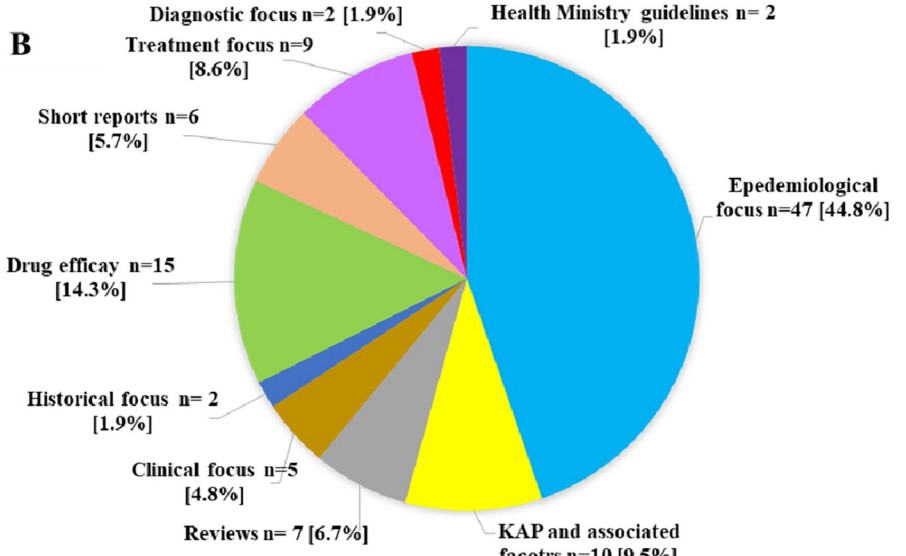

**Fig 2. Results of full-text screening.** (A) Sri Lankan publication productivity on HINIs (B) Number of articles according to their scope. KAP, knowledge, attitude and practices.

## Studies with an epidemiological focus

Of the 47 epidemiological studies, 4 (9%) were islandwide surveys and 43 (91%) were district and provincial-level prevalence assessment studies. Islandwide surveys were conducted in the years 1924, 1937, 2003, and 2017 (Fig 3) involving primary schoolchildren. Province-level prevalence assessment studies were distributed among seven provinces (Fig 4A) and no individual studies were conducted in NWP and NCP. WP was the most assessed province with 15 (34.8%) studies conducted. (Fig 4A). Four studies were conducted concurrently involving several provinces, with two studies focusing on both the WP and SBP, another involving SBP and the CP, and a third in SBP, UP, and the CP. The majority of the epidemiological studies (n = 45, 95.7%) were conducted in community settings while only 2 (4.3%) were hospital-based (Fig 4C). The most sought communities were the general community (n = 21, 46.6%), plantation sector (n = 14, 31.1%) and slum communities (n = 7, 15.5%). Other studies focused on some special communities like indigenous, displaced residing in refugee camps, and

**Table 1. Treatment efficacy studies on HINIs conducted in Sri Lanka.**

| Drug | Dose | Year | Population | *Ascaris* | | *Trichuris* | | Hookworm | | Ref. |
|------|------|------|-----------|-----------|------|-------------|------|----------|------|------|
| | | | | N | ECR% | N | ECR% | N | ECR% | |
| Albendazole | 400mg OD | 1991 | Children | 68 | 95.6 | 85 | 31.8 | 7 | 100 | [47] |
| | | 1999 | Children | - | - | 55 | 43.6 | - | - | [54] |
| Albendazole (Zentel) | 400mg OD | 1996 | Children | 71 | 97.2 | 84 | 26.2 | 59 | 77.9 | [48] |
| Mebendazole | 100mg BD/3 days | 1975 | Children | 62 | 95 | 100 | 70 | 36 | 100 | [45] |
| | | 1990 | General | 95 | 95 | 100 | 79 | 72 | 76 | [49] |
| | 200mg OD | 1991 | Children | 61 | 93.4 | 73 | 37 | 2 | 100 | [47] |
| | 500mg OD | 1987 | Children | 21 | 99 | 14 | 69 | 6 | 87 | [52] |
| | | 1990 | General | 69 | 88 | 91 | 19 | 68 | 19 | [49] |
| | | 1991 | Children | 77 | 97.4 | 94 | 36.2 | 10 | 90 | [47] |
| | | 2013 | General | - | - | - | - | 70 | 28.3 | [56] |
| | 1: 1 mixture of mebendazole polymorph A and C 500mg OD | 2013 | General | - | - | - | - | 74 | 18.8 | [56] |
| Mebendazole (SPMC) | 500mg OD | 1996 | Children | 95 | 95.8 | 110 | 29.1 | 87 | 28.7 | [48] |
| Mebendazole (Janssen) | 500mg OD | 1996 | Children | 84 | 97.6 | 88 | 26.1 | 67 | 35.8 | [48] |
| Flubendazole | 100mg OD/2 days | 1987 | Children | 5 | 100 | 9 | 22 | 7 | 43 | [52] |
| | 200mg OD | 1984 | Children | 47 | 89.4 | 47 | 19.1 | - | - | [51] |
| | 300mg OD | 1987 | Children | 22 | 96 | 40 | 33 | 23 | 78 | [52] |
| | 300mg OD/2 days | 1987 | Children | 7 | 100 | 13 | 46 | 7 | 43 | [52] |
| | 500mg OD | 1984 | Children | 43 | 86 | 43 | 25.8 | - | - | [51] |
| Levamisole | 2.5 mg/kg | 1991 | Children | 73 | 86.3 | 89 | 18 | 8 | 87.5 | [47] |
| Pyrantel pamoate | 10mg/kg OD | 1975 | Adults | - | - | - | - | 10 | 30 | [44] |
| | 10mg/kg OD | 1991 | Children | 68 | 94.1 | 84 | 22.6 | 10 | 90 | [47] |
| | 20mg/kg OD | 1975 | Adults | - | - | - | - | 10 | 40 | [44] |
| | 10mg/kg TD | 1975 | Adults | - | - | - | - | 19 | 73.7 | [44] |
| | 20mg /kg TD | 1975 | Adults | - | - | - | - | 18 | 77.7 | [44] |
| Bephenium hydroxynaphthoate | 5mg OD | 1975 | Adults | - | - | - | - | 10 | 30 | [44] |
| | 5mg TD | 1975 | Adults | - | - | - | - | 19 | 63.1 | [44] |
| Trichloroethylene | 4ml OD | 1975 | Adults | - | - | - | - | 18 | 83.3 | [44] |
| Albendazole -Ivermectin | 400mg or 200μg/kg | 1999 | Children | - | - | 47 | 29.8 | - | - | [54] |
| Albendazole-Diethylcarbamazine | 400mg or 6mg/kg | 1999 | Children | - | - | 53 | 79.3 | - | - | [54] |
| Pyrantel-Oxantel | 20mg/kg | 1987 | Children | 46 | 99 | 43 | 70 | 14 | 94 | [55] |
| Pyrantel-Oxantel -Flubendazole | 20mg/kg or 200mg BD | 1987 | Children | 18 | 100 | 12 | 92 | 2 | 100 | [55] |

OD, single daily dose; BD, two times a day; TD, three times a day; N, number of positive individuals; ECR, egg clearance rate; SPMC, state pharmaceuticals manufacturing corporation

inmates of psychiatric institutions. The majority (80.9%) of the studies were conducted among schoolchildren (Fig 4C). Studies have used different diagnostic methods like DWS, KK, MIFC, formalin ether concentration techniques (FECT), Stoll's egg counting method, culture methods, flotation methods and Scotch tape method. Some studies have used multiple diagnostic methods (Fig 4C). Three STH infections ascariasis, trichuriasis and hookworm infection prevalence were extensively assessed, with 40 (85.1%) on ascariasis, 40 (85.1%) on hookworm infections, and 37 (78.7%) on trichuriasis. Enterobiasis and strongyloidiasis were less frequently assessed with 13 (27.6%), and 8 (17.0%) focusing on their prevalence respectively. Some studies provided prevalence figures for all HINIs, whereas others only provided values for one or two species (Fig 4B).

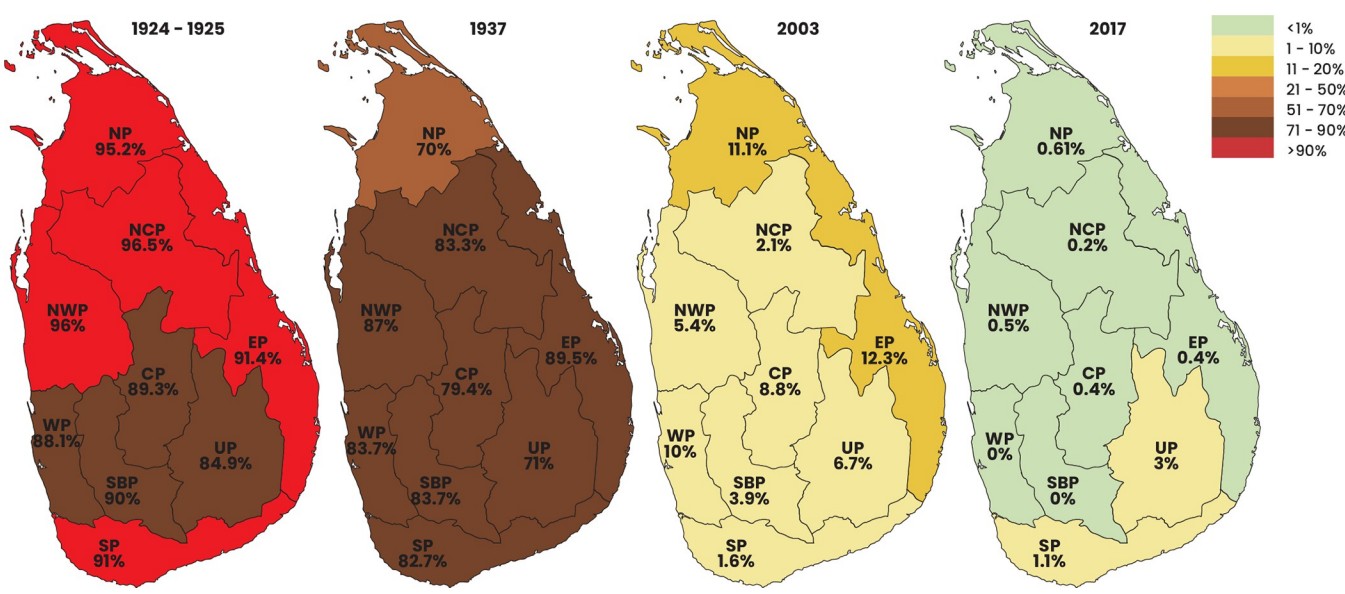

**Fig 3. Cumulative soil-transmitted helminth prevalence in islandwide surveys.** Demarcations within the country are the provincial boundaries. Prevalence value for each province is given. NP, Northern Province; NCP, North Central Province; CP, Central Province; NWP, North Western Province; EP, Eastern Province; UP, Uva Province; SBP, Sabaragamuwa Province; WP, Western Province; SP, Southern Province. (The base layer of the map was sourced from d-maps.com:free maps. https://d-maps.com/carte.php?num_car=56323&lang=en).

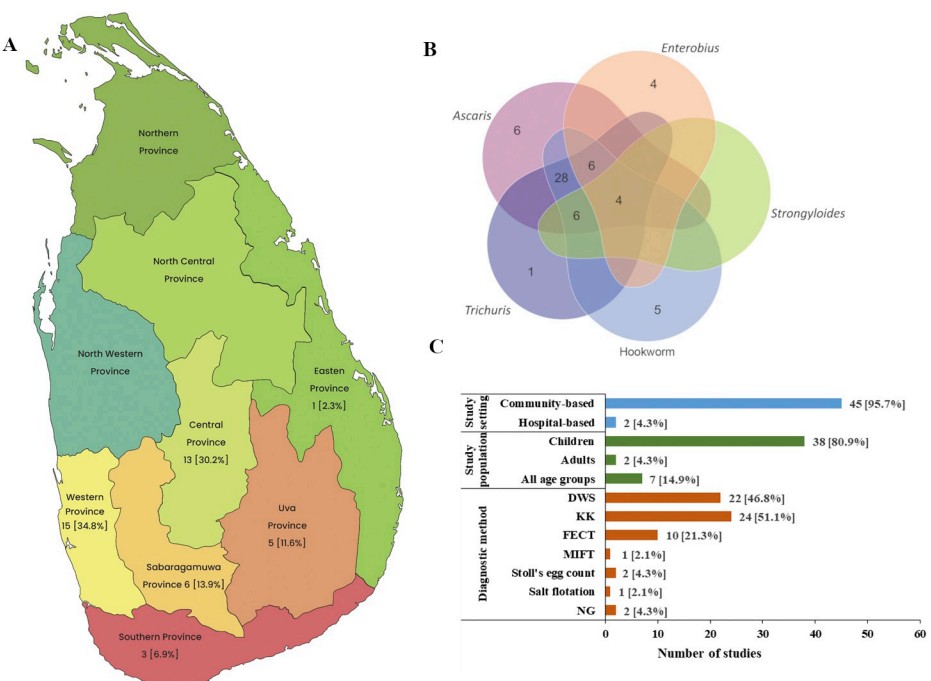

**Fig 4. Epidemiological studies on human intestinal nematode infections in Sri Lanka.** (A) Province-wide distribution of epidemiological studies. Demarcations within the country are the provincial boundaries and the values indicate the number of epidemiological studies carried out in each province. (B) Venn diagram showing different types of human intestinal nematodes assessed in epidemiological studies. (C) Number of epidemiological studies conducted according to their setting, population and diagnostic methods used. DWS, direct wet smear; MIFC, merthiolate iodine formaldehyde concentration; FECT, formalin ether concentration technique; KK, Kato-Katz. (The base layer of the map was sourced from d-maps.com:free maps. https://d-maps.com/carte.php?num_car=56323&lang=en).

## Prevalence and communities affected

The review revealed that HINIs were widely dispersed in Sri Lanka before the year 2000 but gradually declined thereafter. Infections were documented in nearly every geographical and ecological region of the island. During the colonial regime, the highest prevalence rate (98.1%) was reported from the Matale district of the CP [57]. As of today, the Nuwara-Eliya district in the central highlands and the Colombo district in the western lowlands are the most endemic areas for HINIs, with roundworms being the most commonly reported [20].

For the last two decades, twenty-eight prevalence assessment studies have been conducted employing copromicroscopic methods (Table 2). All of them were descriptive cross-sectional studies or national surveys with a minimum sample size of 103 in NP and an islandwide survey having the highest sample size of 5500 [58]. Many studies have focused on WP and CP. Studies show that HINIs are seen among all age groups involving different communities. The majority (n = 24, 85.7%) of studies have assessed the prevalence of at least one of the three main STH infections (ascariasis, trichuriasis, and hookworm infection).

We included twenty-eight prevalence assessment studies conducted over the past two decades for the pooled prevalence assessment (Fig 5). Among a total of 16407 children who participated, involving all the community groups; 2186 were infected with at least one of the three HINIs (*Ascaris*, *Trichuris*, and hookworm), yielding a pooled prevalence of 13.3% (95% CI 12.7–28.9). Of the participants, 1664, 393, and 580 were positive for ascariasis, trichuriasis, and hookworm infections with a pooled prevalence of 10.1% (95% CI 7.6–24.6), 2.4% (95% CI 0.7–5.1) and 3.5% (95% CI 0.1–7.7) respectively.

For the enterobiasis prevalence assessment, 2132 children participated. Of the participants, 698 were positive for enterobiasis giving a pooled prevalence of 32.73% (95% CI 13.77–45.67). Not enough data was available to assess the pooled prevalence of enterobiasis for each population category.

In general community studies involving 11,850 participants, 852 were found positive for at least one STH, resulting in an overall prevalence of 6.96% (95% CI 3.5–30.9). The pooled prevalence rates for ascariasis, trichuriasis, and hookworm infestation were 4.9% (95% CI 1.7–12.3), 1.91% (95% CI 0.4–7.7), and 0.80% (95% CI 0.4–2.7), respectively (Fig 6).

Within the plantation sector community comprising 4,351 children, 1,453 were identified as positive for at least one type of STH infection, resulting in a pooled prevalence of 33.4% (95% CI 26.1–35.1). The pooled prevalence rates for *Ascasis*, *Trichuris*, and hookworm infections were 25.3% (95% CI 21.3–34.5), 4.9% (95% CI 1.1–5.96), and 5.7% (95% CI 1.6–8.3), respectively. Only one study was conducted in the slum community involving 206 children with a cumulative STH prevalence of 11.6% and *Ascaris*, *Trichuris* and hookworm infection prevalence at 9.7%, 1% and 1% respectively (Fig 6). Pooled prevalence assessments for special community groups, such as indigenous and displaced populations, were not conducted due to the limited availability of studies, with only a single study had been performed among these communities.

When examining, several studies did not report on infection intensity. Among the studies that reported infection intensity, the methods of intensity classification were inconsistent, while some reported the mean egg per gram (EPG) count for each species, and others categorised intensity into varying levels such as light, moderate, and heavy, light versus moderate to heavy and light, moderate to heavy and heavy. Due to these inconsistencies and the lack of comprehensive data, we were unable to conduct a thorough assessment of the prevalence of different infection intensities.

## The trends of reported infections

Throughout the review period, a general downward trend in infection prevalence of STHs was observed across all communities studied. However, certain communities, notably those in the

**Table 2. Summary of prevalence assessment studies on human intestinal nematode infections expand in Sri Lanka for the last two decades.**

| No | Author and reference | Study population | Study community | Sample size | Geographical location | Diagnostic method | Prevalence | | | | |
|---|---|---|---|---|---|---|---|---|---|---|---|
| | | | | | | | Hw | Al | Tt | Ss | Ev |
| 1 | Fernando et al (2000) [59] | Primary school children | General community | 743 | Moneragala district, UP | Single KK | 5 | 2 | 0.7 | - | - |
| 2 | Fernando et al (2001) [60] | Primary school children | General community | 349 | Moneragala district, UP | DSS | 1.7 | 0.2 | 0 | - | - |
| 3 | Selvaratnam et al (2003) [61] | Women (19-58 yrs) | Plantation community | 248 | Nuwara Eliya district, CP | Single KK | 10.1 | 31.8 | 10.9 | - | - |
| 4 | De Silva et al (2003) [62] | Primary school children | General community | 265 | Gampaha district, WP | Single KK | 0.7 | 0.7 | 4.1 | - | 0 |
| 5 | Chandrasena et al (2004) [63] | Children 1-3 yrs | Indigenous community | 145 | Badulla district, UP | Single KK, HMT | 20.3 | 0 | 0 | - | 3.1 |
| 6 | Gunawardena et al (2004) [64] | 2–74 yrs old people | Plantation community | 477 | Kegalle district, SBP/Colombo district, WP | Single KK | 28.5 | 52.6 | 67.5 | - | - |
| 7 | Pathmeswaran et al (2005) [65] | Primary school children | General community | 2162 | Islandwide survey | Single KK | 1.2 | 2.8 | 4 | - | - |
| 8 | Banneheka et al (2006) [66] | Primary school children | Plantation community | 316 | Rathnapura district, SBP | DSS, APC, Single KK | 8 | 19 | 2.2 | 0.9 | - |
| 9 | Chandrasena et al (2007) [67] | Children 2–15 yrs | Displaced community | 159 | Vavunia refugee camp, NP | Single KK | 18.2 | 1.3 | 1.3 | - | 0.6 |
| 10 | Gunawardena et al (2008) [68] | Primary school children | General community | 451 | Colombo, Gampaha and Kalutara districts, WP | Single KK | 0.2 | 4.4 | 14.6 | - | - |
| 11 | Chandrasena et al (2010) [69] | All age groups-Women | Inmates of a psychiatric institution | 145 | Colombo district, WP | DSS, Single KK | 0 | 21.4 | 24.8 | 0 | 0 |
| 12 | Kumarendran (2010) [70] | Primary school children | General community | 377 | Nuwara Eliya district, CP | Single KK | - | 36.9 | 3.9 | - | - |
| 13 | Gunawardena et al. (2010) [71] | Primary school children | Plantation community | 1513 | Badulla district, UP/Kandy district, CP/Kegalle and Ratnapura districts, SBP | Single KK | 5.9 | 21.3 | 6.4 | - | - |
| 14 | Gunawardena et al (2011) [72] | School children | Plantation community | 1546 | Nuwara Eliya and Kandy districts, CP/Rathnapura and Kegalle districts, SBP/Badulla district, UP | Single KK | 5.9 | 24.4 | 4.7 | - | - |
| 15 | Rathnayaka et al (2012) [73] | Primary school children | General community | 470 | Badulla district, UP | FECT, Single KK | - | 54 | - | - | - |
| 16 | Karunaithas et al (2012) [58] | Primary school children | General community | 103 | Jaffna district, NP | DSS, IS, FECT and salt flotation | 0 | 0 | 0 | 0 | - |
| 17 | Gunawardane et al (2013) [56] | Primary school children | General community | 260 | Gampaha district, WP | Scotch Tape | - | - | - | - | 38.1 |
| 18 | Gunawardena et al (2013) [74] | Primary school children | General community | 483 | Hambantota district, SP | Scotch Tape | - | - | - | - | 6.2 |
| 19 | Gunawardena et al (2014) [75] | Primary school children | General community | 1882 | Gampaha district, WP | Single KK | 0.3 | 0.3 | 1.3 | | |
| 20 | Suraweera et al (2015) [76] | Children 1–12 yrs | Plantation community | 204 | Kandy district, CP | Scotch Tape | - | - | - | - | 31.9 |
| 21 | Galgamuwa et al, (2016) [77] | Children 1–6 yrs | Plantation community | 254 | Kandy district, CP | DSS, IS, FECT, Single KK | - | 37.8 | - | - | - |
| 22 | Kumarendran et al (2017) [78] | Children 3–7 yrs | General community | 1185 | Colombo district, WP | Scotch Tape | - | - | - | - | 42.5 |
| 23 | Galgamuwa et al (2017) [79] | Children 1–5 yrs. | Slum community | 206 | Kandy district, CP | DSS, IS, FECT, Single KK | 1 | 9.7 | 1 | - | 1 |
| 24 | Lepper et al. (2018) [80] | All age group | Plantation community | 477 | Kegalle district, SBP/Colombo district, WP | Single KK | 28.9 | 52.6 | 67.5 | 0 | 0 |
| 25 | Ubhayawardana et al (2018) [81] | Primary school children | General community | 156 | Colombo district, WP | DSS | 0 | 0 | 0 | 0 | |

*(Continued)*

**Table 2.** (Continued)

| No | Author and reference | Study population | Study community | Sample size | Geographical location | Diagnostic method | Prevalence | | | | |
|---|---|---|---|---|---|---|---|---|---|---|---|
| | | | | | | | Hw | Al | Tt | Ss | Ev |
| 26 | Galgamuwa et al (2018) [82] | Children 1–12 yrs | Plantation Community | 489 | Kandy district, CP | DSS, IS, FECT, Single KK | - | 38.4 | - | - | - |
| 27 | Suraweera et al (2018) [83] | Children 1-12 yrs | Plantation community | 233 | Kandy district, CP | DSS, IS, Single KK | 0 | 26.6 | 0.9 | - | - |
| 28 | Ediriweera et al (2019) [20] | Primary school children | General community | 5500 | Islandwide survey | Duplicate KK | 0.3 | 0.4 | 0.2 | - | - |

Primary school children (grade1-5); yrs, years; DSS, direct saline smear; IS, iodine smear; FECT, formalin ether concentration technique; KK, Kato-Katz; Hw, hookworm; Al, *Ascaris lumbricoides*; Tt, *Trichuris trichiura*; Ss, *Strongyloides stercoralis*; Ev, *Enterobius vermicularis*; NP, Northern Province; NCP, North Central Province; CP, Central Province; NWP, North Western Province; EP, Eastern Province; UP, Uva Province; SBP, Sabaragamuwa Province; WP, Western Province; SP, Southern Province

plantation sector and slums, continue to exhibit higher prevalence rates compared to the general population. According to the latest national survey, conducted in 2017, the cumulative STH prevalence was recorded at 0.97% for the general community, 2.7% for slum areas, and

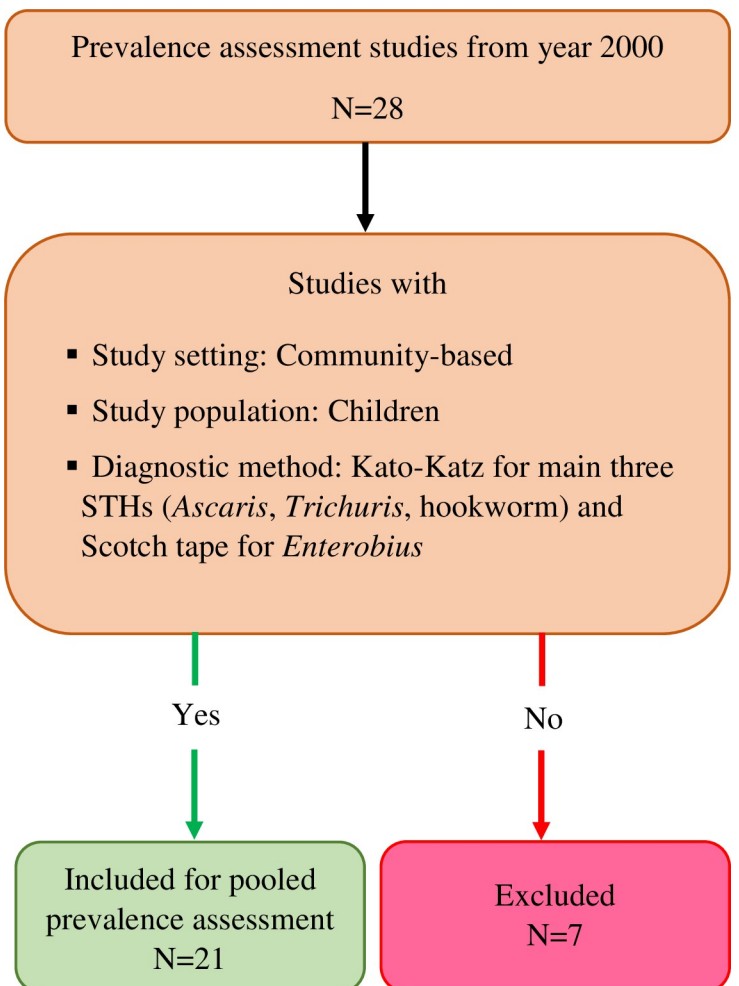

**Fig 5. Studies included in the pooled prevalence assessment.** N, number of studies; STH, soil-transmitted helminths.

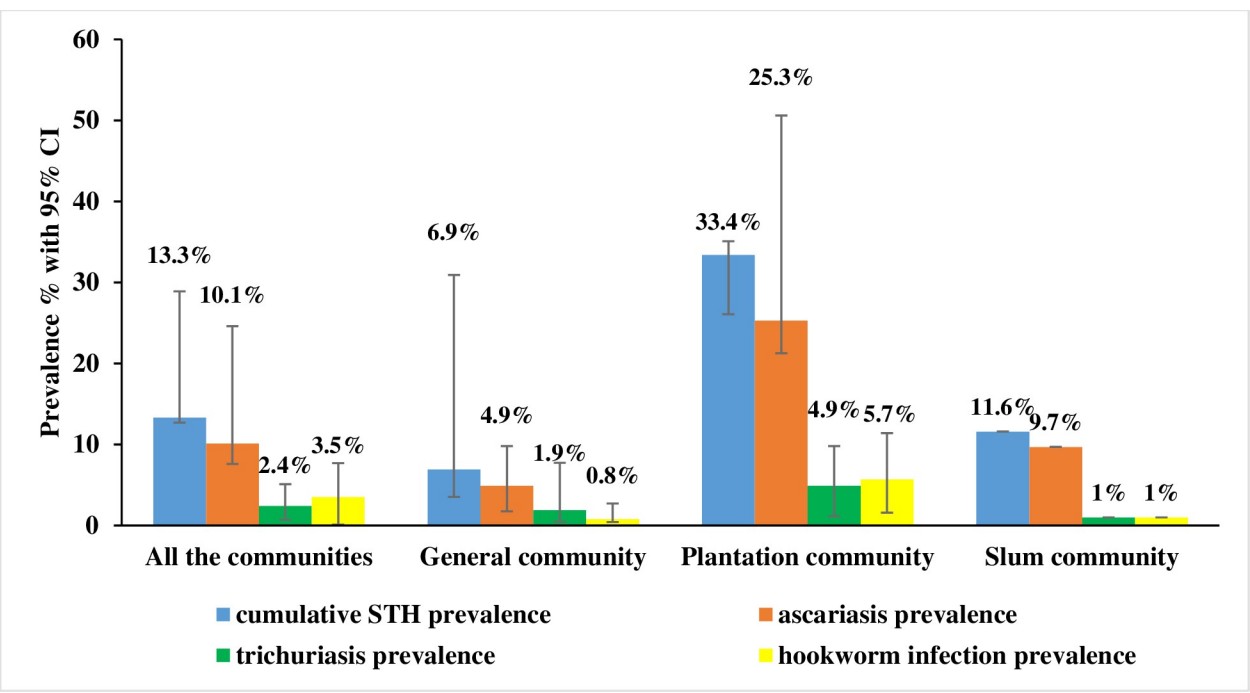

**Fig 6. The pooled prevalence of soil-transmitted helminths (with a 95% confidence interval) among different population categories.** STHs, soil-transmitted helminths.

9.02% for plantation communities [20]. Despite the overall decline in STH infection prevalence, studies assessing *Enterobius* infection consistently revealed high prevalence rates ranging from 30% to 40% [76]. Even the most recent study conducted in 2017 reported a prevalence of 42% [78]. Conversely, *Strongyloides* infections have shown consistently low prevalence rates throughout the review period, with three consecutive studies reporting 0% prevalence. Additionally, prevalence values obtained from special communities such as indigenous (23%), refugees (21%) and inmates of psychiatric institutions (46%) consistently indicate higher trends compared to the general community [63,69,84]. This highlights the importance of targeted interventions and tailored public health strategies to address the specific needs of these vulnerable populations.

Sri Lanka maintained a biannual deworming program targeting school children in the plantation sector from 1994 to 2005 due to an initial STH prevalence of 90% [18]. This was abandoned due to lack of funding and in 2009 study done in the same area showed an increase in prevalence from 19% in 2005 to 29% [66,72]. The study highlights that without sustained preventive measures, infection rates can rebound, emphasising the need for ongoing interventions.

### Factors associated with transmission

Studies have shown that the prevalence of HINIs in any area is determined by factors such as temperature, rainfall, soil, vegetation, drainage, irrigation, coprophagy of domestic animals, sanitation, occupation, and the habits and customs of the people [57]. A plantation industry survey showed that 50.0% of the participants were excreting *Ascaris* eggs [85]. Only 30.7% of the respondents had access to latrines, and nearly all (96.6%) lived in terraces of one-room dwellings, showing that unsanitary living circumstances and subpar sanitation facilities

increase the spread of infection [85]. Geohelminthic infection prevalence was high among a community living in an area where soil contamination was high with geohelminthic eggs [86]. The use of unprotected wells, the absence of water-sealed latrines, and bathing and washing in rock pools created by waterfalls all greatly increase the likelihood of hookworm infections [63]. Poor environmental sanitation, inadequate personal hygiene, limited toilet facilities, unplanned home garden cultivation, and poverty were found to be variables contributing to the prevalence of HINIs in the communities. Over time, improvements in sanitary facilities in the country may have had a favourable impact on the decline in HINI prevalence. All KAP surveys showed that knowledge among participants was poor [86–88]. This could be a reason that some of the communities continue to have relatively high levels of HINI prevalence despite the control measures.

## Preventive and control measures

Population growth in Sri Lanka necessitated large-scale investment in WASH. Between 2015 and 2019, there was a notable increase in budgetary expenditure for WASH, reaching the Sri Lankan rupee (LKR) 49.5 billion in 2019, reflecting the economic development and the government's growing commitment [89]. However, during the COVID-19 pandemic, the emphasis on sanitation increased, diverting attention from other disease control measures. Even though 90% of households have access to safe drinking water and sanitation, strong spatial differences exist with only 36% having access to piped water and 2% to piped sewerage [89]. Coverage is more in urban areas with 99.9% of the population in Colombo having access to clean water while it is 54% in Nuwara Eliya where many of the plantation sector communities reside with high levels of HINIs reported [90]. Sri Lanka aims to provide universal access to safe water by 2025, sanitation by 2030 and end open defecation by 2025 [89]. To bridge decades of uneven progress with improved access to water and sanitation and to achieve the above targets the country needs a well-planned government policy [91]. Sri Lanka has undertaken several completed and ongoing projects, including the Gift Water Project, Access to Water for Communities Affected by Drought (AWCAD) Project, Water & Sanitation in Schools and Communities (WSSC) Project and WASH-TE II Project, to contribute towards these aims [92]. Government and health authorities have taken continuous steps with medications to improve and maintain a low infection prevalence in the country.

In the early 20th century, Sri Lanka pursued mass deworming programs. From 1994 to 2005, a biannual program targeting school-aged children with mebendazole addressed the high prevalence of STH infections [18]. However, after its cessation due to funding issues, a 2009 study showed a high overall prevalence of STH infections (29.0%) in estate sector schools, indicating a rebound effect [18]. For the first time in 2012, the Family Health Bureau (FHB) of Sri Lanka's Ministry of Health issued guidelines for community-based deworming of children and pregnant women [18]. These guidelines categorised regions into high-risk, moderate-risk and low-risk areas, based on national surveys. They recommended biannual deworming with mebendazole for children in high-risk areas and annual deworming for those in moderate-risk regions [93]. Pregnant women were also included in the program due to the adverse effects of hookworm infections, particularly on iron deficiency anaemia [93]. The national survey done in 2017 showed a decline in the overall prevalence of STH infections across the country, prompting the FHB to revise the deworming guidelines issued by the Ministry of Health by discontinuing deworming in low-risk areas and tailing down the deworming in other areas.

## Regional comparisons

When comparing Sri Lanka's HINI situation with other countries, notable differences and similarities emerge. India has the highest microscopic prevalence of STH infections in South

Asia, which is 21% (13). With an overall prevalence of 18%, *Ascaris* was the most common species in South Asia, followed by *Trichuri*s (14%) and hookworm (12%) (13). Bangladesh showed a notable reduction in STH prevalence from 79.8% in 2005 to 14% in 2020, attributed to the introduction of the MDA program in 2008 [94]. China successfully reduced STH prevalence from 53.58% in 1988–1992 to 0.84% in 2020 [95]. Countries with robust healthcare systems and extensive deworming programs, such as Singapore and Japan, have successfully maintained low STH prevalence levels [90]. Similarly, in Africa, countries like Ethiopia and Nigeria face challenges akin to economic instability and limited access to healthcare exacerbating STH prevalence [96,97]. Studies conducted in Myanmar reveal that while the microscopic prevalence of STH is 33.3%, the molecular prevalence is significantly higher at 78% [98]. There was no comprehensive data on the molecular prevalence of HINIs for many countries, except for some regions. For example, in Vietnam, molecular studies have revealed the unequal distribution of STH infection among different areas implying the STH control programmes may not be reaching certain areas. However, more research is needed to fully understand the molecular epidemiology of STH infections in these countries and its impact on control efforts.

One of the problems with STH prevalence assessment studies in Asia is the broader variety of diagnostic techniques utilised making it challenging to compare the outcomes directly [13]. Many countries have implemented regular deworming programs, resulting in reduced prevalence of STH infections. However, certain regions within these countries continue to experience high prevalence rates due to dense populations and inadequate sanitation facilities. While most nations conduct mass deworming initiatives targeting schoolchildren, some have extended these efforts to include adults [99]. Integrated control programs, such as the antifilarial and deworming program in Sri Lanka and the African Programme for Onchocerciasis Control (APOC), demonstrate the efficacy of collaborative approaches in reducing government costs associated with deworming [99]. Regional collaborations, such as the East Asia Summit (EAS) and the Association of Southeast Asian Nations (ASEAN), provide platforms for sharing experiences and best practices in STH control, facilitating efforts to reduce the burden of STH infections across Asia [100]. These comparisons underscore the importance of contextual factors, healthcare infrastructure, and collaborative strategies in addressing the complexity of STH infections.

## Discussion

For the first time, the current study presents an in-depth account of the trends of infection, current prevalence, associated factors, diagnostic methods, prevention and control strategies of HINIs in Sri Lanka. This scoping review is timely, coinciding with Sri Lanka's achievement of a prevalence of less than 1% for STH and cessation of regular deworming in three districts, in alignment with the WHO's strategy to eliminate STH by 2030 and reduce the required number of tablets in preventive chemotherapy [20]. Over the past 20 years, the STH prevalence in Sri Lanka remained low and continued to decline gradually reaching 0.97% in 2017 [7,20] Throughout the period, slum communities and plantation sector communities showed a higher prevalence of HINIs compared to the general community remaining at 2.73% and 9.02% respectively in 2017 [20]. This can be attributed to the socioeconomic disparities, crowded living conditions and low education levels seen in these areas compared to other regions of the country [89]. The implementation, continuation and monitoring of regular school-based deworming programmes positively impacted the control of infection in Sri Lanka [93].

There is a notable gap in research from regions like NWP, NCP, EP, SBP, SP and NP, especially in recent years and at-risk populations with frequent soil exposure occupations. Less

attention was given to studying intestinal nematodes like *E. vermicularis* and *S. stercoralis*. The WHO 2021–2030 roadmap for neglected tropical diseases (NTDs) recommends establishing an effective strongyloidiasis control program for school-age children [101]. However, in the Sri Lankan context, it is crucial to first have a comprehensive understanding of the prevalence of strongyloidiasis within the community through extensive surveillance efforts. Prevalence assessment studies have not incorporated molecular methods, which offer greater sensitivity, particularly considering the low infection intensity prevalent in the country. This could potentially be due to the high initial costs, maintenance challenges, scarcity of skilled personnel, infrastructure limitations, inconsistent supply chains, and expensive consumables, which pose significant challenges in a resource-limited setting like Sri Lanka [102,103]. Moreover, many studies rely on single stool samples for analysis and non-gold standard techniques such as DWS, FECT, and KK were used to assess strongyloidiasis and enterobiasis prevalence. These factors collectively contribute to the likelihood of underreporting HINI prevalence.

Incorporating enterobiasis prevalence assessment into the national survey would be beneficial, given its high prevalence in conducted studies. This inclusion could lead to more effective control strategies. KAP surveys highlight low awareness and misconceptions regarding symptoms, transmission, prevention, and deworming [88]. Therefore, prioritising health promotion programs to enhance knowledge and improve attitudes and practices is crucial. Furthermore, addressing overcrowding and inadequate infrastructure facilities in slum and plantation sector communities should be a government priority at all administrative levels to effectively control these issues. As the last national survey gives recommendations up to 2022, conducting another survey is important, especially in regions where deworming was discontinued. Molecular prevalence assessment will benefit in accurately identifying HINI prevalence in the country.

This study has several limitations. An in-depth analysis of the prevalence was difficult as most of the studies were without adequate methodological detail with regard to the subjects and the diagnostic techniques. For example, a broader variety of diagnostic techniques utilised, such as DWS, FECT, KK and salt flotation techniques with different sensitivities and specificities prevents a reasonable comparison of the outcomes. As this scoping review aims to explore Sri Lanka's HINI history, distribution, trends, control, diagnostics, and challenges, we avoided constraining the time frame, or specific publication types except for case reports, nor did we limit studies based on stringent methodological criteria. Consequently, encompassing a broader timeframe and, period incorporating much grey literature lacking in methodological detail rendered, a thorough meaningful analysis of the quality of evidence provided by the publications was unfeasible. We only assessed the methodological quality of the prevalence assessment studies incorporated for the pooled prevalence analysis.

As HINIs are endemic in many regions worldwide, lessons learned from Sri Lanka's experiences can inform policy decisions and intervention strategies in other countries facing similar challenges. The identification of regional disparities and the evaluation of control strategies offer actionable insights for policymakers and public health practitioners working to achieve targets set forth by organisations. By fostering collaboration among researchers, policymakers, healthcare providers, and community stakeholders, this review shows the multisectoral approach necessary for achieving sustainable improvements in public health outcomes worldwide. By advocating for equitable access to healthcare, education, and socioeconomic opportunities, stakeholders can address the underlying determinants of HINIs and other NTDs, ultimately working towards health equity on a global scale. In summary, this scoping review on HINIs in Sri Lanka not only contributes valuable insights to the national context but also holds significant implications for global health. By informing policies, advancing research agendas, fostering collaboration, and advocating for equity, this review contributes to the collective effort to eliminate NTDs and improve health outcomes for populations worldwide.

## Conclusion

The efforts to eliminate HINIs in Sri Lanka have led to the classification of certain districts as low, moderate, or high risk in line with the WHO risk classification. Despite numerous studies conducted in some districts and communities, we observed a low number of published research on HINIs in general and a persistent vulnerability among slum and plantation communities. Children, pregnant women, slum dwellers, and plantation workers remain most susceptible, with roundworm and hookworm being the most prevalent infections. While prevalence has generally declined, strict comparisons between studies are hindered by methodological differences. Sri Lanka's successful control programs offer valuable lessons for other countries in the region as well as across the globe, emphasising comprehensive approaches including health education, sanitation, treatment, and community engagement. Adaptation to local contexts is key to success in combating these infections worldwide.

## Supporting information

**S1 PRISMA Checklist. PRISMA-ScR checklist.** Preferred Reporting Items for Systematic reviews and Meta-Analyses extension for Scoping Reviews Checklist.
(PDF)

**S1 Table. Quality assessment of epidemiological studies from the year 2000, based on the modified Newcastle-Ottawa Quality Assessment Scale**
(PDF)

## Acknowledgments

We would like to thank Ms Chithra M Abeygunasekera, Senior Assistant Librarian, Faculty of Medicine, University of Kelaniya for her valuable support.

## Author Contributions

**Conceptualization:** Nalini Kaushalya Jayakody, Kosala Gayan Weerakoon.

**Data curation:** Nalini Kaushalya Jayakody, Nilanthi de Silva, Kosala Gayan Weerakoon.

**Formal analysis:** Nalini Kaushalya Jayakody, Nilanthi de Silva, Kosala Gayan Weerakoon.

**Funding acquisition:** Kosala Gayan Weerakoon.

**Methodology:** Nalini Kaushalya Jayakody, Anjana Silva, Susiji Wickramasinghe, Nilanthi de Silva, Sisira Siribaddana, Kosala Gayan Weerakoon.

**Project administration:** Nalini Kaushalya Jayakody, Kosala Gayan Weerakoon.

**Resources:** Nalini Kaushalya Jayakody, Anjana Silva, Susiji Wickramasinghe, Nilanthi de Silva, Sisira Siribaddana, Kosala Gayan Weerakoon.

**Software:** Nalini Kaushalya Jayakody, Nilanthi de Silva, Kosala Gayan Weerakoon.

**Supervision:** Anjana Silva, Susiji Wickramasinghe, Nilanthi de Silva, Sisira Siribaddana, Kosala Gayan Weerakoon.

**Validation:** Nalini Kaushalya Jayakody, Anjana Silva, Susiji Wickramasinghe, Nilanthi de Silva, Sisira Siribaddana, Kosala Gayan Weerakoon.

**Visualization:** Nalini Kaushalya Jayakody, Kosala Gayan Weerakoon.

**Writing – original draft:** Nalini Kaushalya Jayakody.

**Writing – review & editing:** Nalini Kaushalya Jayakody, Anjana Silva, Susiji Wickramasinghe, Nilanthi de Silva, Sisira Siribaddana, Kosala Gayan Weerakoon.

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
