## [Decision Letter · Decision Letter 0]

29 Aug 2024

Dear Dr Jayakody,

Thank you very much for submitting your manuscript "Human intestinal nematode infections in Sri Lanka: a scoping review" for consideration at PLOS Neglected Tropical Diseases. As with all papers reviewed by the journal, your manuscript was reviewed by members of the editorial board and by independent reviewers. In light of the reviews (below this email), we would like to invite the resubmission of a significantly-revised version that takes into account the reviewers' comments. 

The reviewers have raised several comments that require adequate responses. Accordingly, appropriate corrections and modifications must be made in the manuscript to be considered for publication.

We cannot make any decision about publication until we have seen the revised manuscript and your response to the reviewers' comments. Your revised manuscript is also likely to be sent to reviewers for further evaluation.

Sincerely,

Timir Tripathi, Ph.D.

Academic Editor

Jong-Yil Chai

Section Editor

The reviewers have raised several comments that require adequate responses. Accordingly, appropriate corrections and modifications must be made in the manuscript to be considered for publication.

Reviewer's Responses to Questions

**Key Review Criteria Required for Acceptance?**

**Methods**

-Are the objectives of the study clearly articulated with a clear testable hypothesis stated?

-Is the study design appropriate to address the stated objectives?

-Is the population clearly described and appropriate for the hypothesis being tested?

-Is the sample size sufficient to ensure adequate power to address the hypothesis being tested?

-Were correct statistical analysis used to support conclusions?

-Are there concerns about ethical or regulatory requirements being met?

Reviewer #1: To a large extent yes, but comments on the methods are below

Reviewer #2: Yes

Reviewer #3: Conducted according to properly defined methodology accepted for a scoping review followed by appropriate analyses.

**Results**

-Does the analysis presented match the analysis plan?

-Are the results clearly and completely presented?

-Are the figures (Tables, Images) of sufficient quality for clarity?

Reviewer #1: Yes, well done in this section.

Reviewer #2: Yes

Reviewer #3: Results are clearly presented.

**Conclusions**

-Are the conclusions supported by the data presented?

-Are the limitations of analysis clearly described?

-Do the authors discuss how these data can be helpful to advance our understanding of the topic under study?

-Is public health relevance addressed?

Reviewer #1: Yes

Reviewer #2: Yes

Reviewer #3: Limitations are identified. Discussion has highlighted broader relevance to similar settings and policy decisions.

**Editorial and Data Presentation Modifications?**

Reviewer #1: It was a pleasure reviewing this work. The work was well done and reflected in the rigor and findings. I recommend discretionary accept with minor revisions.

sincerely

Reviewer #2: (No Response)

Reviewer #3: Line 80 - infected to be replaced with 'infective'

**Summary and General Comments**

Reviewer #1: Thank you for the opportunity to review your work entitled, “ Human intestinal nematode infections in Sri Lanka: a scoping review,” which explores the distribution, trends, diagnostics, control and challenges of HINI in Sri Lanka.

While I commend the efforts of the authors in bringing this study to bear, I have a few comments and concerns for considerations to potentially improve the quality and clarity of your work. Please find below.

Abstract

Your introduction section sets a good stage for reader engagement starting with the setting of Sri Lanka, the burden of disease of HINI, the gaps in literature informing the study objectives. Welldone.

More could be mentioned in the methodology section here on how the data were synthesized from the data sources, and perhaps the keywords/query search strategy to warrant a systematic scoping methodology.

Also, how many articles were identified for inclusions before diving into the findings.

“HINIs in Sri Lanka were first described in 1888 in the annual survey report of the Principal Civil Medical Officer.” This is already established as a burden in the introduction. Its presence here adds no value to the objectives -i.e. distribution, trends, diagnostics, control and challenges of HINI in Sri Lanka.

I commend you describe your findings too to reflect the objectives of your inquiry. Same goes for your conclusions.

Overall your abstract section should tell the key details of a PRISMA-SCr guided report done in a way that engages readers to want to dive in more into your study.

Introduction.

Enjoyed reading this section. Pay attention to repetitions and redundancies. Also confirm your claims are cited per author guidelines.

Methods

“... adhering to the Preferred Reporting Items for Systematic Reviews and Meta-Analyses extension for Scoping Reviews (PRISMA-Scr).” I don’t think this statement is accurate. The frameworks were built upon one another as described but the PRISMA-Scr was designed by Tricco and colleagues as a reporting guideline and not a framework that Arksey and co needed to adhere to. 

The 5 steps you identified are Arksey and O’Malley’s reporting formats and not the PRISMA-Scr reporting format. I will advise you pick one and stick to it rather than conflate both as you did.

Did I miss the section on ensuring rigor, quality assurance in your article selection?

What about critical appraisals and evaluation of included articles? What that done? If not why?

Reviewer #2: This is an interesting paper in an important area.

General comments

In this paper, you only provided data on the prevalence of any intensity of infections, which can be used to stop preventive chemotherapy when this prevalence is less than 2%. However, to declare the elimination of HINIs as a public health problem, defined as the prevalence of moderate-to-heavy intensity of infections less than 2%. There is no reported data on that in your manuscript. Can you provide this? In cases where there is no data, you mentioned it in the text.

Comments

Line 33: You mentioned the end period of the search, which was September 2022. Can you provide the starting period?

Line 495: You mentioned less attention was given to studying intestinal nematodes like E. vermicularis and S. stercoralis without further recommendation. Could you add a recommendation to include S. stercoralis in preventive as recommended by WHO in its new roadmap over 2021 – 2030 for the control of soil-transmitted helminths?

Line 496: You stated that prevalence assessment has not incorporated molecular methods that offer greater sensitivity. Could you provide more details on that aspect? Is that a feasibility issue in terms of cost-efficiency compared to the traditional diagnostic method?

Reviewer #3: A pertinent review which highlights trends of common intestinal helminth infections in a background of improving living conditions.

PLOS authors have the option to publish the peer review history of their article (what does this mean?). If published, this will include your full peer review and any attached files.

Reviewer #1: No

Reviewer #2: No

Reviewer #3: No
---

## [Decision Letter · Decision Letter 1]

8 Nov 2024

Dear Dr Jayakody,

We are pleased to inform you that your manuscript 'Human intestinal nematode infections in Sri Lanka: a scoping review' has been provisionally accepted for publication in PLOS Neglected Tropical Diseases.

Best regards,

Timir Tripathi, Ph.D.

Academic Editor

Jong-Yil Chai

Section Editor

Shaden Kamhawi

co-Editor-in-Chief

Paul Brindley

co-Editor-in-Chief

Reviewer's Responses to Questions

**Key Review Criteria Required for Acceptance?**

**Methods**

-Are the objectives of the study clearly articulated with a clear testable hypothesis stated?

-Is the study design appropriate to address the stated objectives?

-Is the population clearly described and appropriate for the hypothesis being tested?

-Is the sample size sufficient to ensure adequate power to address the hypothesis being tested?

-Were correct statistical analysis used to support conclusions?

-Are there concerns about ethical or regulatory requirements being met?

Reviewer #1: Yes the are. The authors paid attention to reviewer comments and revised the methods to reflect that

Reviewer #2: Yes

**Results**

-Does the analysis presented match the analysis plan?

-Are the results clearly and completely presented?

-Are the figures (Tables, Images) of sufficient quality for clarity?

Reviewer #1: Yes

Reviewer #2: Yes

**Conclusions**

-Are the conclusions supported by the data presented?

-Are the limitations of analysis clearly described?

-Do the authors discuss how these data can be helpful to advance our understanding of the topic under study?

-Is public health relevance addressed?

Reviewer #1: Very well

Reviewer #2: Yes

**Editorial and Data Presentation Modifications?**

Reviewer #1: (No Response)

Reviewer #2: (No Response)

**Summary and General Comments**

Reviewer #1: No further comments or issues

Reviewer #2: The author addressed all my comments

PLOS authors have the option to publish the peer review history of their article (what does this mean?). If published, this will include your full peer review and any attached files.

Reviewer #1: **Yes: **Udoka Okpalauwaekwe

Reviewer #2: No

---

## [Editor Report · Acceptance letter]

25 Nov 2024

Dear Dr Jayakody,

We are delighted to inform you that your manuscript, "Human intestinal nematode infections in Sri Lanka: a scoping review," has been formally accepted for publication in PLOS Neglected Tropical Diseases.

Best regards,

Shaden Kamhawi

co-Editor-in-Chief

Paul Brindley

co-Editor-in-Chief
